# WHY DO THESE MATCH? EXPLAINING THE BEHAVIOR OF IMAGE SIMILARITY MODELS

## ABSTRACT

Explaining a deep learning model can help users understand its behavior and allow researchers to discern its shortcomings. Recent work has primarily focused on explaining models for tasks like image classification or visual question answering. In this paper, we introduce an explanation approach for image similarity models, where a model's output is a score measuring the similarity of two inputs rather than a classification. In this task, an explanation depends on both of the input images, so standard methods do not apply. We propose an explanation method that pairs a saliency map identifying important image regions with an attribute that best explains the match. We find that our explanations provide additional information not typically captured by saliency maps alone, and can also improve performance on the classic task of attribute recognition. Our approach's ability to generalize is demonstrated on two datasets from diverse domains, Polyvore Outfits and Animals with Attributes 2.

## 1 INTRODUCTION

Many problems in artificial intelligence that require reasoning about complex relationships can be solved by learning some feature embedding to measure similarity between images and/or other modalities such as text. Examples of these tasks include scoring fashion compatibility (Han et al., 2017b; Hsiao & Grauman, 2018; Vasileva et al., 2018), image retrieval (Kiapour et al., 2015; Radenovi et al., 2018; Yelamarthi et al., 2018), or zero-shot recognition (Bansal et al., 2018; Li et al., 2018b; Wang et al., 2018). Reasoning about the behavior of similarity models can aid researchers in identifying potential improvements, show where two images differ for anomaly detection, promote diversity in fashion recommendation by ensuring different traits are most prominent in the top results, or simply help users understand the model's predictions which can build trust (Teach & Shortliffe, 1981). However, prior work on producing explanations for neural networks has primarily focused on explaining classification models (*e.g.* (Fong & Vedaldi, 2017; Nguyen et al., 2016; Petsiuk et al., 2018; Ribeiro et al., 2016; Selvaraju et al., 2017; Zeiler & Fergus, 2014)) and does not directly apply to similarity models. Given a *single* input image, such methods produce a saliency map which identifies pixels that played a significant role towards a particular class prediction (see Figure 1a for an example). On the other hand, a similarity model requires at least *two* images to produce a score. The interaction between both images defines which features are more important, so replacing just one of the images can result in identifying different salient traits.

Another limitation of existing work is that the saliency alone may be insufficient as an explanation of (dis)similarity. For image pairs where similarity is determined by the presence or absence of an object, a saliency map may be enough to understand model behavior. However, when we consider the image pair in Figure 1b, highlighting the necklace as the region that contributes most to the similarity score is reasonable, but uninformative given that there are no other objects in the image. Instead, what is important is the fact that the necklace shares a similar color with the ring. Whether these attributes or salient parts are a better fit as an explanation is not determined by the image domain (*i.e.* attributes for e-commerce imagery vs. saliency for natural imagery), but instead by the images themselves. For example, an image can be matched as formal-wear because of a shirt's collar (salient part), while two images of animals can match because both have stripes (attribute).

Guided by this intuition, we introduce *Salient Attributes for Network Explanation (SANE)*. Our approach generates a saliency map to explain a model's similarity score, paired with an attribute explanation that identifies important image properties. SANE is a "black box" method, meaning it

Figure 1: Existing explanation methods focus on image classification problems (left), whereas we explore explanations for image similarity models (right). We pair a saliency map, which identifies important image regions, but often provides little useful information, with an attribute (*e.g.*, golden), which is more human-interpretable and, thus, a better explanation than saliency alone.

can explain any network architecture and only needs to measure changes to a similarity score when provided with different inputs. Unlike a standard classifier, which simply predicts the most likely attributes for a given image, our explanation method predicts which attributes are important for the similarity score predicted by a model. Predictions are made for each image in a pair, and allowed to be non-symmetric, *e.g.*, the explanation for why the ring in Figure 1b matches the necklace may be that it contains "black", even though the explanation for why the necklace matches the ring could be that it is "golden." A different similarity model may also result in different attributes being deemed important for the same pair of images.

SANE combines three major components: an attribute predictor, a prior on the suitability of each attribute as an explanation, and a saliency map generator. Our underlying assumption is that at least one of the attributes present in each image should be able to explain the similarity score assigned to the pair. Given an input image, the attribute predictor outputs a confidence score and activation map for each attribute, while the saliency map generator produces regions important for the match. During training, SANE encourages overlap between the similarity saliency and attribute activation. At test time, we rank attributes as explanations for an image pair based on a weighted sum of this attribute-saliency map matching score, the explanation suitability prior of the attribute, and the likelihood that the attribute is present in the image. Although we evaluate only the top-ranked attribute in our experiments, in practice more than one attribute could be used to explain a similarity score. We find that using saliency maps as supervision for the attribute activation maps during training not only improves the attribute-saliency matching, resulting in better attribute explanations, but also boosts attribute recognition performance using standard metrics like average precision.

We evaluate several candidate saliency map generation methods which are primarily adaptations of "black box" approaches that do not rely on a particular model architecture or require access to network parameters to produce a saliency map (Fong & Vedaldi, 2017; Petsiuk et al., 2018; Ribeiro et al., 2016; Zeiler & Fergus, 2014). These methods generally identify important regions by measuring a change in the output class score resulting from some perturbation of the input image. Similarity models, however, typically rely on a learned embedding space to reason about relationships between images, where proximity between points or the lack thereof indicates some degree of correspondence. An explanation system for embedding models must, therefore, consider how distances between embedded points, and thus their similarity, change based on perturbing one or both of the input images. We explore two strategies for adapting these approaches to our task. First, we manipulate just a single image (the one we wish to produce an explanation for) while keeping the other image fixed. Second, we manipulate both images to allow for more complex interactions between the pair. See Section 3.2 for additional details and discussion on the ramifications of this choice.

Our paper makes the following contributions: 1) we provide the the first quantitative study of explaining the behavior of image similarity models; 2) we propose a novel explanation approach that combines saliency maps and attributes; 3) we validate our method with metrics designed to link our explanations to model performance, and find that it produces more informative explanations than adaptations of prior work to this task and also improves attribute recognition performance.

## 2 RELATED WORK

**Saliency-based Explanations.** Saliency methods can generally be split into "white box" and "black box" approaches. "White box" methods assume access to internal components of a neural network,

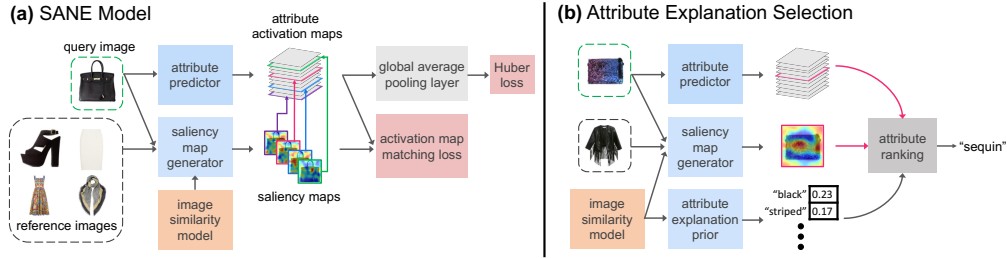

Figure 2: Approach Overview. During training, each saliency map produced by the generator is encouraged to match at least one ground truth attribute's activation maps. Then, at test time, we rank attribute explanations by how well the saliency and attribute activation maps match, along with the likelihood of the attribute and its explanation suitability prior. Note that we assume the image similarity model has been pretrained and is kept fixed in all our experiments

either in the form of gradients or activations of specific layers (*e.g.* (Cao et al., 2015; Chang et al., 2019; Nguyen et al., 2016; Selvaraju et al., 2017; Simonyan et al., 2014; Yosinski et al., 2015; Zhang et al., 2016; Zhou et al., 2016)). Most of them produce a saliency map by using some version of backpropagation from class probability to an input image. In contrast, "black box" approaches require no knowledge of the internals (*e.g.* weights, gradients) of the models. These methods obtain saliency maps by perturbing the input in a predefined way and measuring the effect of it on the model output, such as class score. We adapt and compare three "black box" and one "white box" methods for our saliency map generator in Figure 2. "Black box" approaches include a Sliding Window (Zeiler & Fergus, 2014), which masks image regions sequentially, and Randomized Input Sampling for Explanations (RISE) (Petsiuk et al., 2018), which masks random sets of regions. Both measure the effect removing these regions have on the class score. LIME (Ribeiro et al., 2016) first obtains a super-pixel representation of an image. Super-pixel regions are randomly deleted, and their importance is estimated using Lasso. "White box" Mask (Fong & Vedaldi, 2017) learns a saliency map directly by using different perturbation operators and propagating the error to a low resolution mask. Although there is some limited work which adapts some saliency methods to the image similarity setting (*e.g.*, Gordo & Larlus (2017)), they present qualitative results only, *i.e.*, these methods are not evaluated quantitatively on their explanation accuracy as done in our work.

**Natural Language and Attribute-based Explanations.** Instead of producing saliency maps, which can sometimes be difficult to interpret, researchers have explored methods of producing text-based explanations. These include methods which justify a model's answer in the visual question answering task (Huk Park et al., 2018; Li et al., 2018a), rationalize the behavior of a self-driving vehicle (Kim et al., 2018b), or describe why a category was selected in fine-grained object classification (Hendricks et al., 2016). Lad & Parikh (2014) used human-generated attribute explanations describing why two images are similar or dissimilar as guidance for image clustering. Our approach could be used to automatically generate these explanations rather than relying on human feedback. Several works exist which learn attribute explanations either to identify important concepts (Kim et al., 2018a; Bau et al., 2017; Fong & Vedaldi, 2018) or to justify a model's decision by pointing to evidence (Anne Hendricks et al., 2018). Kim et al (2018a) learns a concept activation vector that separates examples with an attribute against examples without it, and then scores the sensitivity of attributes based on how often a directional derivative changes the inputs towards the concept. However, all these methods were designed to explain categorical predictions rather than similarity models. To the best of our knowledge, ours is the first work which uses attribute explanations in the image similarity setting.

## 3   SALIENT ATTRIBUTES FOR NETWORK EXPLANATIONS (SANE)

We are given a fixed model that predicts the similarity between two images, and must explain why a query image is similar to a reference image. While typical models for predicting similarity are learned from data, using an embedding method and a triplet loss, our approach is agnostic as to how the model being explained is built. Our method consists of three components: the attribute explanation model (Section 3.1), the saliency map generator (Section 3.2), and an attribute explanation suitability prior (Section 3.3). Although we train a CNN to produce attribute predictions, the image

similarity model we wish to explain is kept fixed. At test time, one recovers a saliency map for the match from the query image in a pair, then uses the attribute explanation model and attribute suitability prior to rank each attribute's ability to explain the image similarity model.

## 3.1 ATTRIBUTE EXPLANATION MODEL

Suppose we have access to pairs of images $(I_r, I_q)$. Here, $I_r$ denotes a reference image and $I_q$ a query image. We wish to obtain an explanation for the match between $I_r$ and $I_q$. Associated with each pair is a saliency map $\mathbf{m}_q$ produced by a saliency map generator (described in Section 3.2). We need simply to swap the query and reference images to compute a saliency map for $\mathbf{m}_r$, which would also likely result in a different saliency map than $\mathbf{m}_q$. Finally, assume we have access to binary attribute annotations $a_i$, $i = 1, \ldots, A$, and let $\mathbf{a}_{gt} \in \{0, 1\}^A$ be the set of ground truth attribute annotations for a given query image. If no attribute annotations are provided, an attribute discovery method could be employed (*e.g.*, (Han et al., 2017a; Vittayakorn et al., 2016)). We explore using an attribute discovery method in the appendix.

Our attribute explanation model produces confidence scores $\hat{\mathbf{a}} \in \mathbb{R}^A$ for $I_q$. Unlike a standard attribute classifier, however, our goal is not just to predict the most likely attributes in $I_q$, but rather to identify which attributes contribute the most to the similarity score $s(I_r, I_q)$ produced by the similarity model we wish to obtain explanations for. To accomplish this, the layer activations for attribute $a_i$ before the global average pooling layer is defined as an attribute activation map $\mathbf{n}_i$. This attribute activation map represents a downsampled mask of an image that identifies prominent regions in $I_q$ for that attribute. We encourage at least one ground truth attribute's activation map for image $I_q$ to match saliency map $\mathbf{m}_q$ as a form of regularization (see Figure 2 for an overview). Our underlying assumption is that at least one of the ground truth attributes of $I_q$ should be able to explain why $I_q$ is similar to $I_r$. Thus, at least one of the attribute activation maps $\mathbf{n}_i$ should closely resemble the saliency map for the match, $\mathbf{m}_q$.

Each attribute confidence score is obtained using a global average pooling layer on its attribute activation map followed by a softmax. The attribute explanation network is trained using a Huber loss (Huber, 1964), sometimes referred to as a smooth $\ell_1$ loss, which helps encourage sparsity in the predictions. More formally, given a set of confidence scores $\hat{\mathbf{a}}$ and attribute labels $\mathbf{a}_{gt}$, our loss is,

$$L_{Huber}(\hat{\mathbf{a}}, \mathbf{a}_{gt}) = \begin{cases} \frac{1}{2}(\mathbf{a}_{gt} - \hat{\mathbf{a}})^2 & \text{for } |\mathbf{a}_{gt} - \hat{\mathbf{a}}| \leq 1 \\ \mathbf{a}_{gt} - \hat{\mathbf{a}} & \text{otherwise.} \end{cases} \tag{1}$$

Note that multiple attributes can be present in the image; and that this loss operates on attributes, not attribute activation maps. Since the confidence scores sum to one (due to the softmax), we scale a binary label vector by the number of ground truth attributes $A_{gt}$ (*e.g.*, if there are four attributes for an image, its label would be 0.25 for each ground truth attribute, and zero for all others).

**Leveraging saliency maps during training.** We explicitly encourage our model to identify attributes which are useful in explaining the predictions of an image similarity model by finding which attributes best describe the regions of high importance to similarity predictions. To accomplish this, we first find a set of regions that may be important to the decisions of an image similarity model by generating a set of $K$ saliency maps $\mathcal{M}_q$ for up to $K$ reference images that are similar. For the image under consideration, we also construct a set of attribute activation maps $\mathcal{N}_{gt}$ corresponding to each ground truth attribute. Then, for each saliency map we find its best match in $\mathcal{N}_{gt}$. We match saliency maps to attributes rather than the other way around since not all annotated attributes are necessarily relevant to the explanation of $s(I_r, I_q)$. We use an $\ell_2$ loss between the selected attribute activation map and saliency map, *i.e.*,

$$L_{hm} = \frac{1}{K} \sum_{\forall \mathbf{m} \in \mathcal{M}_q} \min_{\forall \mathbf{n} \in \mathcal{N}_{gt}} \|\mathbf{m} - \mathbf{n}\|_2 . \tag{2}$$

Combined with the attribute classification loss, our model's complete loss function is:

$$L_{total} = L_{Huber} + \lambda L_{hm}, \tag{3}$$

where $\lambda$ is a scalar parameter. See appendix for implementation details and parameter values.

### 3.2 SALIENCY MAP GENERATOR

A straightforward method of producing a saliency map with a "black box" method is to manipulate the input image by removing image regions and measuring its effect on the similarity score. If a large drop in similarity is measured, then the region must be significant. If almost no change was measured, then the model considers the image region irrelevant. A saliency map is generated from this approach by averaging the similarity scores for each pixel over all instances where it was removed from the input. The challenge is determining the best way of manipulating the input image to discover these important regions. These "black box" methods assume you don't have access to the underlying model's parameters to compute gradients, and as our experiments will show, can also produce saliency maps that are often comparable or better than some "white box" methods which leverage these gradients. We compare four saliency methods: Sliding Window (Zeiler & Fergus, 2014), LIME (Ribeiro et al., 2016), "white box" Mask (Fong & Vedaldi, 2017), and RISE (Petsiuk et al., 2018). We now describe how we adapt these models for our task; additional details on each method are in the appendix.

**Computing similarity scores**. Each saliency method we compare was designed to operate on a single image and measures the effect manipulating the image has on the prediction of a specific object class. However, an image similarity models predictions are made for two or more images. Let us consider the case where we are just comparing two images, a query image (*i.e.* the image we want to produce an explanation for), and a reference image, although our approach extends to consider multiple reference images. Even though we do not have access to a class label, we can measure the effect manipulating an image has on the similarity score between the query and reference images. Two approaches are possible: manipulate both images, or manipulate only the query image.

**Manipulating both images** would result in $NM$ forward passes through the image similarity model (for $N, M$ the number of query and reference image manipulations, respectively), which is prohibitively expensive unless $M << N$. But we only need an accurate saliency map for the query image, so we set $M << N$ in our experiments. There is another danger: for example, consider two images of clothing items that are similar if either they both contain or do not contain a special button. Masking out the button in one image and not the other would cause a drop in similarity score, but masking out the button in both images would result in high image similarity. These conflicting results could make accurately identifying the correct image regions contributing to a score difficult.

The alternative is to **manipulate the query image** alone, *i.e.* keep a fixed reference image. We evaluate the saliency maps produced by both methods in Section 4.1.

### 3.3 SELECTING INFORMATIVE ATTRIBUTES

At test time, given a similarity model and a pair of inputs, SANE generates a saliency map and selects an attribute to show to the user. We suspect that not all attributes annotated for a dataset may prove to be useful in explaining every image similarity model. Thus, we take into account each attribute's usefulness in explaining predictions made by a similarity model by learning concept activation vectors (CAVs) (Kim et al., 2018a) over the image similarity embedding. These CAVs identify which attributes are useful in explaining a layer's activations by looking at whether an attribute positively affects the model's predictions. CAVs are defined as the vectors that are orthogonal to the classification boundary of a linear classifier trained to recognize an attribute over a layer's activations (*i.e.* the image similarity model's embedding). Then, the sensitivity of each concept to an image similarity model's predictions (*i.e.*, the TCAV score) is obtained by finding the fraction of features that were positively influenced by the concept using directional derivatives computed using a triplet loss with a margin of machine epsilon. Note that this creates a single attribute ranking over the entire image similarity embedding (*i.e.*, it is agnostic to the image pair being explained) which we use as an attribute explanation suitability prior. Finally, attributes are ranked as explanations using a weighted combination of the TCAV scores, the attribute confidence score $\hat{\mathbf{a}}$, and how well the attribute activation map $\mathbf{n}$ matches the generated saliency map $\mathbf{m}_q$. *I.e.*,

$$\mathbf{e}(\mathbf{m}_q, \hat{\mathbf{a}}, \mathbf{n}, \text{TCAV}) = \phi_1 \hat{\mathbf{a}} + \phi_2 \, d_{\cos}(\mathbf{m}_q, \mathbf{n}) + \phi_3 \text{TCAV}, \qquad (4)$$

where $d_{\cos}$ denotes cosine similarity, and $\phi_{1-3}$ are scalars estimated via grid search on held out data.

## 4 EXPERIMENTS

**Datasets.** We evaluate our approach using two datasets from different domains to demonstrate its ability to generalize. The Polyvore Outfits dataset (Vasileva et al., 2018) consists of 365,054 fashion product images annotated with 205 attributes and composed into 53,306/10,000/5,000 train/test/validation outfits. Animals with Attributes 2 (AwA) (Xian et al., 2018) consists of 37,322 natural images of 50 animal classes annotated with 85 attributes, and is split into 40 animal classes for training, and 10 used at test time. To evaluate our explanations we randomly sample 10,000 ground-truth (query, reference) pairs of similar images for each dataset from the test set.

**Image Similarity Models.** For the Polyvore Outfits dataset we use the type-aware embedding model released by Vasileva et al. (2018). This model captures item compatibility (*i.e.* how well two pieces of clothing go together) using a set of learned projections on top of a general embedding, each of which compares a specific pair of item types (*i.e.* a different projection is used when comparing a top-bottom pair than when comparing a top-shoe pair). For AwA we train a feature representation using a 18-layer ResNet (He et al., 2016) with a triplet loss function that encourages animals of the same type to embed nearby each other.[1] For each dataset/model, cosine similarity is used to compare an image pair's feature representations.

### 4.1 SALIENCY MAP EVALUATION

**Metrics.** Following Petsiuk et al. (2018), we evaluate the generated saliency maps using insertion and deletion metrics which measure the change in performance of the model being explained as pixels are inserted into a blank image, or deleted from the original image. For our task, we generate saliency maps for all query images, and insert or delete pixels in that image only. If a saliency map correctly captures the most important image regions, we should expect a sharp drop in performance as pixels are deleted (or a sharp increase as they are inserted). We report the area under the curve (AUC) created as we insert/delete pixels at a rate of 1% per step for both metrics. We normalize the similarity scores for each image pair across these thresholds so they fall in a [0-1] interval.

**Results.** Table 1 compares the different saliency map generation methods on the insertion and deletion tasks. RISE performed best on most metrics, with the exception of and LIME doing better on the deletion metric on AwA. This is not surprising, since LIME learns which super-pixels contribute to a similarity score. For AwA this means that parts of the animals could be segmented out and deleted or inserted in their entirety before moving onto the next super-pixel. On Polyvore Outfits, however, the important components may be along the boundaries of objects (*e.g.* the cut of a dress), something not well represented by super-pixel segmentation. Although Mask does not perform as well as other approaches, it tends to produce the most compact regions of salient pixels as it searches for a saliency map with minimal support (see the appendix for examples). Notably, we generally obtained better performance when the reference image was kept fixed and only the query image was manipulated. This may be due to the issues from noisy similarity scores as discussed in Section 3.2 and suggests extra care must be taken when manipulating both images.

### 4.2 ATTRIBUTE PREDICTION EVALUATION

**Metrics.** To begin, we report the overall performance of SANE using mean average precision (mAP) on the standard task of attribute recognition computed over all images in the test set. Two additional metrics are used to evaluate our attribute explanations using the (query, reference) image pairs used in the saliency map experiments which are similar to the evaluation of saliency maps. Given the set of attributes we know exist in the image, we select which attribute among them best explains the similarity score using Eq. (4), and then see the effect of deleting that attribute from the image has on the similarity score. Similarly, we select the attribute which best explains the similarity score from those which aren't present in the image and measure the effect of inserting the attribute has on the similarity score. Intuitively, if an attribute was critical for an explanation, then the similarity score should shift more than if a different attribute was selected. Scores for these metrics are expressed in terms of relative difference, so higher numbers are always better regardless of whether we are inserting or deleting an attribute. Scores are also normalized so the upper bound is 1.

---

[1]Upon acceptance, we will make our embedding model for the AwA dataset publically available.

Table 1: Comparison of candidate saliency map generator methods described in Section 3.2. We report AUC for the insertion and deletion metrics described in Section 4.1.

| Method | Fixed Reference? | Polyvore Outfits | | Animals with Attributes 2 | |
|---|---|---|---|---|---|
| | | Insertion ($\uparrow$) | Deletion ($\downarrow$) | Insertion ($\uparrow$) | Deletion ($\downarrow$) |
| Sliding Window | Y | 60.2 | 53.6 | 76.9 | 76.8 |
| LIME | Y | 58.4 | 55.4 | 77.0 | **71.2** |
| Mask | Y | 59.4 | 53.3 | 74.5 | 77.3 |
| RISE | Y | **64.3** | **49.5** | 77.4 | 75.1 |
| Sliding Window | N | 59.6 | 54.3 | 77.6 | 76.3 |
| Mask | N | 58.9 | 54.6 | 75.8 | 78.4 |
| RISE | N | 61.4 | 51.9 | **77.7** | 74.0 |

To insert or remove an attribute, we find a representative image in the test set which is most similar to the query image in all other attributes. For example, let us consider a case where we want to remove the attribute "striped" from a query image. We would search through the database for the image which is most similar in terms of non-striped attributes, but which hasn't been labeled as being "striped." These attribute comparisons are based on the average confidence for each attribute computed over the three attribute models we compare in Table 2. On the Polyvore Outfits dataset we restrict the images considered to be of the same type as the query image (*i.e.*, if the query image is a shoe, then only a shoe can be retrieved). After retrieving this new image, we compute its similarity with the reference image to compare with the original (query, reference) image pair. Examples of this process can be found in the appendix.

**Compared methods.** We provide three baseline approaches: a random baseline, a sample attribute classifier (*i.e.* no attribute activation maps), and a modified version of FashionSearchNet (Ak et al., 2018), an attribute recognition model which also creates a weakly-supervised attribute activation map for comparison. Additional details on these models can be found in the appendix.

**Results.** Table 2 compares the performance of the compared attribute models for our metrics. Our attribute explanation metrics demonstrate the effectiveness of our attribute explanations, with our model which matches saliency maps and includes TCAV scores getting the best performance on both datasets. This shows that when we "insert" or "delete" the attribute predicted by SANE from the image, it affects the similarity model's score more than the baselines. Notably, our approach outperforms FashionSearchNet + Map Matching, which can be considered a weakly-supervised version of SANE trained for attribute recognition. The second line of Table 2 reports that TCAV does well on the insertion task, but poorly on the deletion task. This is due, in part, to the fact that other models, including SANE, are trained to reason about attributes that actually exist in an image, whereas for insertion the goal is to predict which attribute that isn't present in the image would affect the similarity score most significantly. Thus, using a bias towards globally informative attributes (*i.e.*, TCAV scores) is more useful for insertion. Finally, training SANE to produce explanations leads to a 2% improvement on the standard attribute recognition task measured in mAP over a simple attribute classifier while using the same number of parameters. SANE also outperforms FashionSearchNet, which treats localizing important image regions as a latent variable rather than using saliency maps for supervision.

We conducted a small scale user study to verify humans found our explanations useful. We showed the users the saliency map explanations for image pairs or the attribute explanations, and asked if they felt the explanations helped them understand the model's behavior. We obtained at least 25 responses for each question, where 56% of subjects found the saliency maps useful and 72% found the attributes useful on the AwA dataset, while on Polyvore Outfits these are 87% and 79%, respectively. This verifies that most users thought our explanations were useful, which could help build trust, while also showing how the different explanations are ideal in different circumstances.

We provide qualitative examples of our explanations in Figure 3. Examples demonstrate that our explanations pass important sanity checks. The explanation attribute is well-correlated with the localization of important pixels in the saliency map for each pair. Notice that "golden", "striped" and "printed" in the first two columns of Figure 3 are sensibly localized, and are also reasonable explanations for the match, while a more abstract explanation like "fashionable" is linked to the high heel, the curve of the sole, and the straps of the shoe. Note further that the explanations are non-trivial: they more often than not differ from the most likely attribute in the query image,

Table 2: Comparison of how attribute recognition (mAP) and attribute explanation (insertion, deletion) metrics described in Section 4.2 are affected for different approaches. We use fixed-reference RISE as our saliency map generator for both datasets. Higher numbers are better for all metrics.

| Method | Polyvore Outfits | | | Animals with Attributes 2 | | |
| --- | --- | --- | --- | --- | --- | --- |
| | mAP | Insertion (↑) | Deletion (↑) | mAP | Insertion (↑) | Deletion (↑) |
| Random | – | 71.7 | 73.3 | – | 41.2 | 48.9 |
| TCAV (Kim et al., 2018a) | – | 75.3 | 74.3 | – | 47.6 | 49.7 |
| Attribute Classifier | 28.8 | 72.8 | 74.8 | 65.1 | 41.9 | 50.0 |
| FashionSearchNet (Ak et al., 2018) | 29.6 | 73.2 | 74.6 | 66.3 | 41.8 | 50.3 |
| FashionSearchNet + Map Matching | – | 72.8 | 75.2 | – | 42.8 | 50.8 |
| SANE | **30.9** | 73.5 | 74.9 | **67.3** | 42.4 | 50.2 |
| SANE + Map Matching | – | 73.7 | 75.7 | – | 44.1 | 51.3 |
| SANE + Map Matching + TCAV (Full) | – | **76.8** | **76.2** | – | **48.9** | **51.6** |

Figure 3: Qualitative results of our attribute explanations for pairs of examples on the Polyvore Outfits and the AwA datasets. The attribute predicted as explanation for each reference-query match is shown below the saliency map. The most likely attribute for the query image as predicted by our attribute classifier is shown directly underneath it.

as predicted by a standard attribute classifier. In other words, our explanation model is utilizing information from each pair of images and the saliency map characterizing the match to produce a sensible, interpretable explanation.

Figure 4 shows an example of how directly removing the attribute predicted as the explanation can affect similarity (possible here because the attribute is a color.) Here we see that when we modify the white dress to be a different color, the similarity score drops significantly. The only exception is when we make the dress the same color (black) as the attribute explanation of the pants it is being compared to. This demonstrates in a causal way how our predicted attributes can play a significant role in the similarity scores.

## 5 CONCLUSION

In this paper we introduced SANE, a method of explaining an image similarity model's behavior by identifying attributes that were important to the similarity score paired with saliency maps indicating import image regions. We confirmed humans believe our explanations are useful for explaining a model's behavior, which could help build trust, to supplement automatic metrics. In future work we believe closely integrating the saliency generator and attribute explanation model, enabling each component to take advantage of the predictions of the other, would help improve performance.

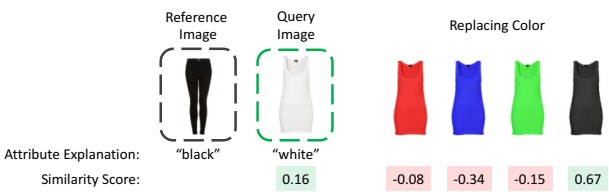

Figure 4: Example of the effect replacing the attribute used as an explanation of the model's behavior has on image similarity score (higher score means items are more compatible).

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

## A  CANDIDATE SALIENCE MAP GENERATOR DESCRIPTIONS

In this section we provide additional details about each of the candidate saliency map generation methods used in our paper. We split these approaches into two groups: methods which analyze behavior solely through input manipulation (described in Section A.1) and those which use an optimization procedure to learn some parameters in combination with input manipulation (described in Section A.2). Please see Section 3.2 of our paper for a description of how these methods are adapted to our task. We also provide a runtime comparison of each approach in Table 3. A qualitative comparison between saliency map generators on the Polyvore Outfits and AwA datasets is provided in Figures 5 and 6.

### A.1 SALIENCY MAPS BY INPUT MANIPULATION

A straightforward approach to producing a saliency map is to manipulate the input image by removing image regions and measuring the effect this has on the similarity score. If a large drop in similarity is measured, then the region must be important to this decision. If almost no change was measured, then the model considers the image region irrelevant. The saliency map is generated from this approach by averaging the similarity scores for each pixel location over all instances where it was removed from the input. The challenge then is to determine how to manipulate the input image to discover these important regions.

**Sliding Window (Zeiler & Fergus, 2014).** The first approach to removing regions of an image we shall discuss is a sliding window, where regions are sampled regularly across an image. There is a direct tradeoff, however, with how densely frames are sampled and the computational time it takes to do a forward pass through the network for each manipulated image. If frames are not densely sampled to enable an efficient solution, then it wouldn't be able to localize important regions accurately. If regions are too densely sampled then removing them might not make enough of a difference in the similarity score to take measurements accurately.

**RISE (Petsiuk et al., 2018).** This method uses Monte Carlo approach to generate saliency maps. A set of $N$ random binary masks of size $h \times w$ is sampled where each element is independently set to 1 with probably $p$, and all other elements are set to 0. Typically these masks are much smaller than the input image, so they are upsampled using bilinear interpolation. This produces small continuous regions within the upsampled mask that can be used to manipulate the input image. To remove the fixed grid structure the masks are upsampled to larger than image size and then cropped randomly. Although this approach does require a significant number of random masks (we found 2,000 to be sufficient in our experiments), we found this approach significantly outperforms using a sliding window that samples a similar number of masks on our task.

### A.2 LEARNED SALIENCY MAPS

We shall now discuss methods which combine input manipulation with an optimization procedure used to directly learn a saliency map. As in Section A.1, we compare generating saliency maps for a single query image at a time using a fixed reference image as well as generating a saliency map by manipulating both the query and reference images.

**LIME (Ribeiro et al., 2016).** Rather than masking regions without any concern over the continuity of a region, this approach to generating saliency maps operates over a superpixel segmentation of an image. Images are manipulated by randomly deleting superpixels in the image. After sampling $N$ manipulated inputs, the importance of each superpixel is estimated using Lasso. Finally, important regions are selected using submodular optimization.

**Mask (Fong & Vedaldi, 2017).** In this approach a low resolution saliency map is directly learned using stochastic gradient decent and upsampled to the image size. Instead of manipulating an image by just deleting regions as in other methods, two additional perturbation operators are defined: adding Gaussian noise and image blurring. To help avoid artifacts when learning the mask a total-variation norm is used in addition to an $L1$ regularization to promote sparsity. This approach removes the reliance on superpixels and tends to converge in fewer iterations than LIME, although it is considerably slower in practice than other approaches (see Table 3). That said - one advantage it does have over other approaches is the ability to learn the salience map for both the query and reference image jointly (which we take advantage of when we are not using a fixed reference image).

## B ADDITIONAL EXPERIMENTAL OR IMPLEMENTATION DETAILS

### B.1 SALIENCY MAP GENERATOR DETAILS

**Sliding Window.** When manipulating the inputs of the reference image, we apply 625 occlusion windows each covering a square region of about 12% of image area. When manipulating both images we apply 36 occlusion windows to the reference image.

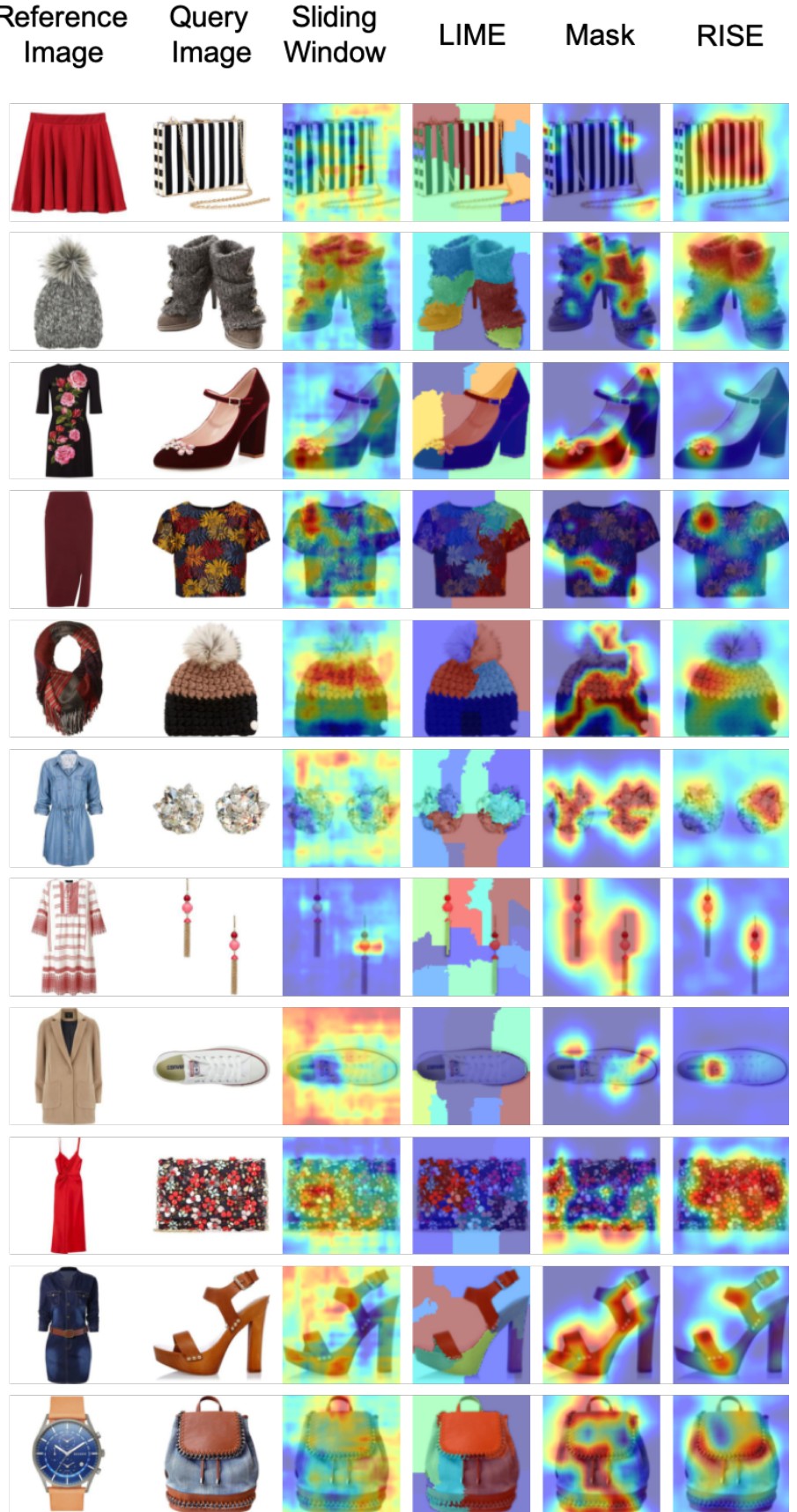

Figure 5: Qualitative examples comparing the saliency map generator candidates on the Polyvore Outfits dataset.

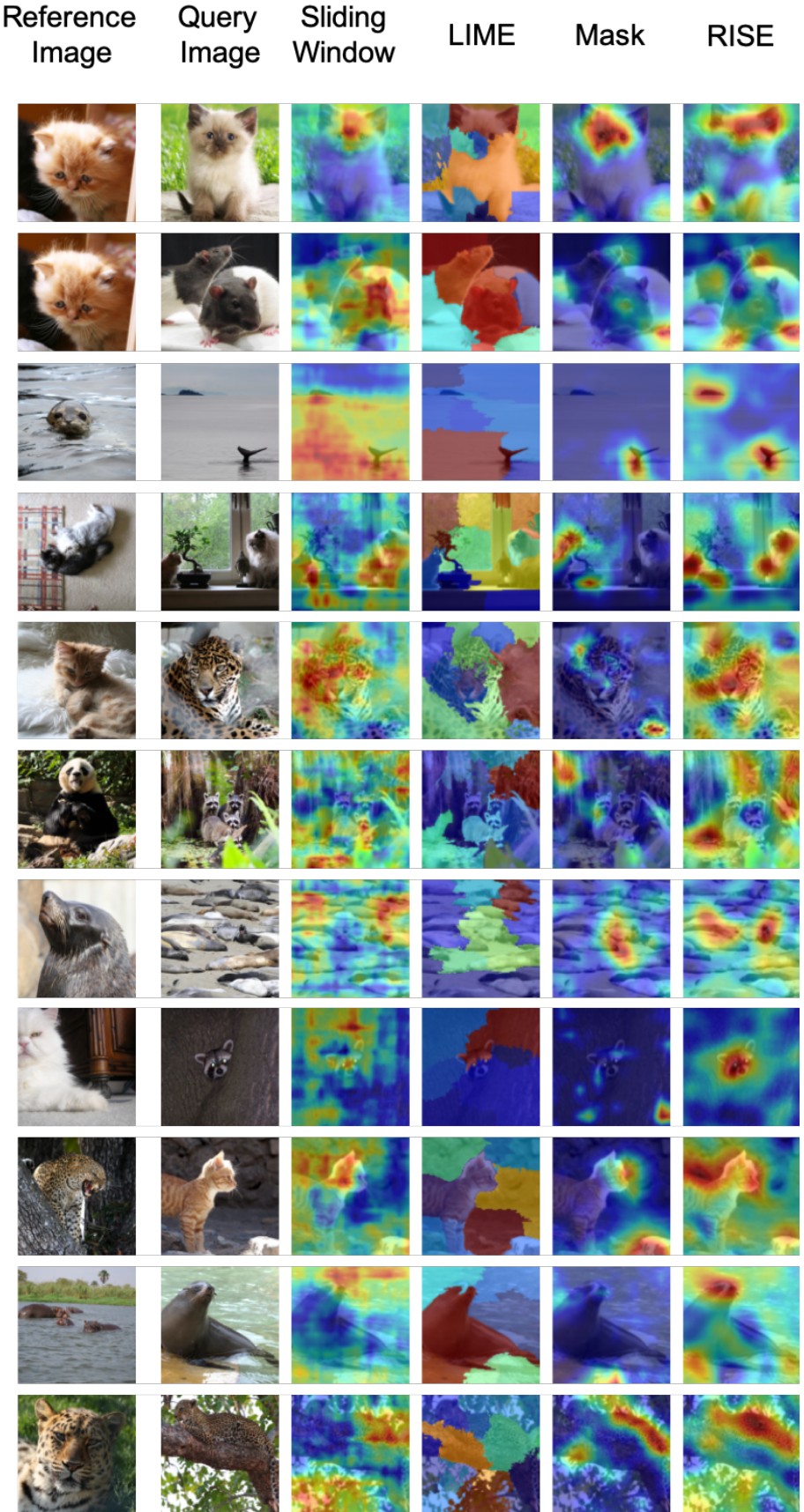

Figure 6: Qualitative examples comparing the saliency map generator candidates on the Animals with Attributes dataset.

Table 3: Runtime comparison of the compared saliency generation methods and how using a fixed reference image, or manipulating both the query and reference images affects performance.

| Method | Fixed Reference? | Time(s) |
|---|---|---|
| Sliding Window | Y | 0.2 |
| LIME | Y | 1.2 |
| Mask | Y | 4.1 |
| RISE | Y | 0.3 |
| Sliding Window | N | 2.5 |
| Mask | N | 7.2 |
| RISE | N | 5.8 |

**RISE.** For both datasets we randomly sample 2,000 random masks upsampled from $8 \times 8$ mask with the probability of preserving a region of $0.5$. When manipulating the inputs of the reference image, we generate 30 random masks.

**LIME.** We generate LIME saliency maps using $1000$ samples.

**Mask.** We learn a $14 \times 14$ perturbation mask for both datasets. We train the mask for 500 iterations using Adam Kingma & Ba (2015) with a learning rate of 0.1.

## B.2 COMPARED ATTRIBUTE METHODS

In addition to a random baseline, we provide two for comparison to our model for our attribute experiments in Section 4.2. First, we train a simple attribute classifier using the same architecture as SANE, but where the attribute activation maps are not regularized with saliency maps. Second, we use a modified version of FashionSearchNet (Ak et al., 2018), which was designed for fashion search using attribute information. This network uses an attribute activation map to identify and extract a region of interest for each attribute. These extracted regions are fed into two branches consisting of three fully connected layers which is trained for both attribute classification and image retrieval. We remove the image retrieval components For all attribute models, we use a 50-layer ResNet base image encoder He et al. (2016) and the last convolutional layer of the network the same number of channels as the number of classes. This provides a simple baseline and a model with a generic weakly-supervised attribute activation map for comparison.

## B.3 SANE DETAILS

Due to its efficient (see Table 3) and overall good performance (see Table 1) we selected the fixed-reference RISE as our saliency map generator. For each training image, we sample up to five similar images using the ground truth annotations of each dataset and generate saliency maps using each sampled image as the reference image. We train our attribute model for 300 epochs using Adam (Kingma & Ba, 2015) with a learning rate of $5e^{-4}$ and set $\lambda = 5e^{-3}$ in Eq. 3 from the paper. After each epoch, we computed mAP on the validation set and kept the best performing model according to this metric. Additional qualitative examples for explanations produced by our SANE model on the Polyvore Outfits and AwA datasets are provided in Figures 7 and 8.

We provide an example of the attribute deletion process in Figure 9. After identifying an attribute to remove in an image, we search for the most similar image to the input from a database that doesn't contain the input attribute. Image similarity is computed over the attribute space, *i.e.*, we want to keep the predictions of each attribute the same, and only vary the target attribute. For Polyvore Outfits, we only consider images of the same type (*i.e.*, so tops can only be replaced with other tops). To ensure we don't bias towards a single attribute model, average the predictions made by each attribute model in our experiments (Attribute Classifier, FashionSearchNet, and SANE). We see on the left side of Figure 9 that some attributes like colors are largely retained when the attribute has to do with a non-color based attribute. On the returned AwA images on the right side of Figure 9 we see how some attributes can lead to significant changes in the images or almost none at all depending on the attribute selected to remove.

In Section 3.3 we discuss how we estimate how likely each attribute is a "good" explanation in held-out data. This is used as a prior to bias our attribute selections towards attributes that are known to

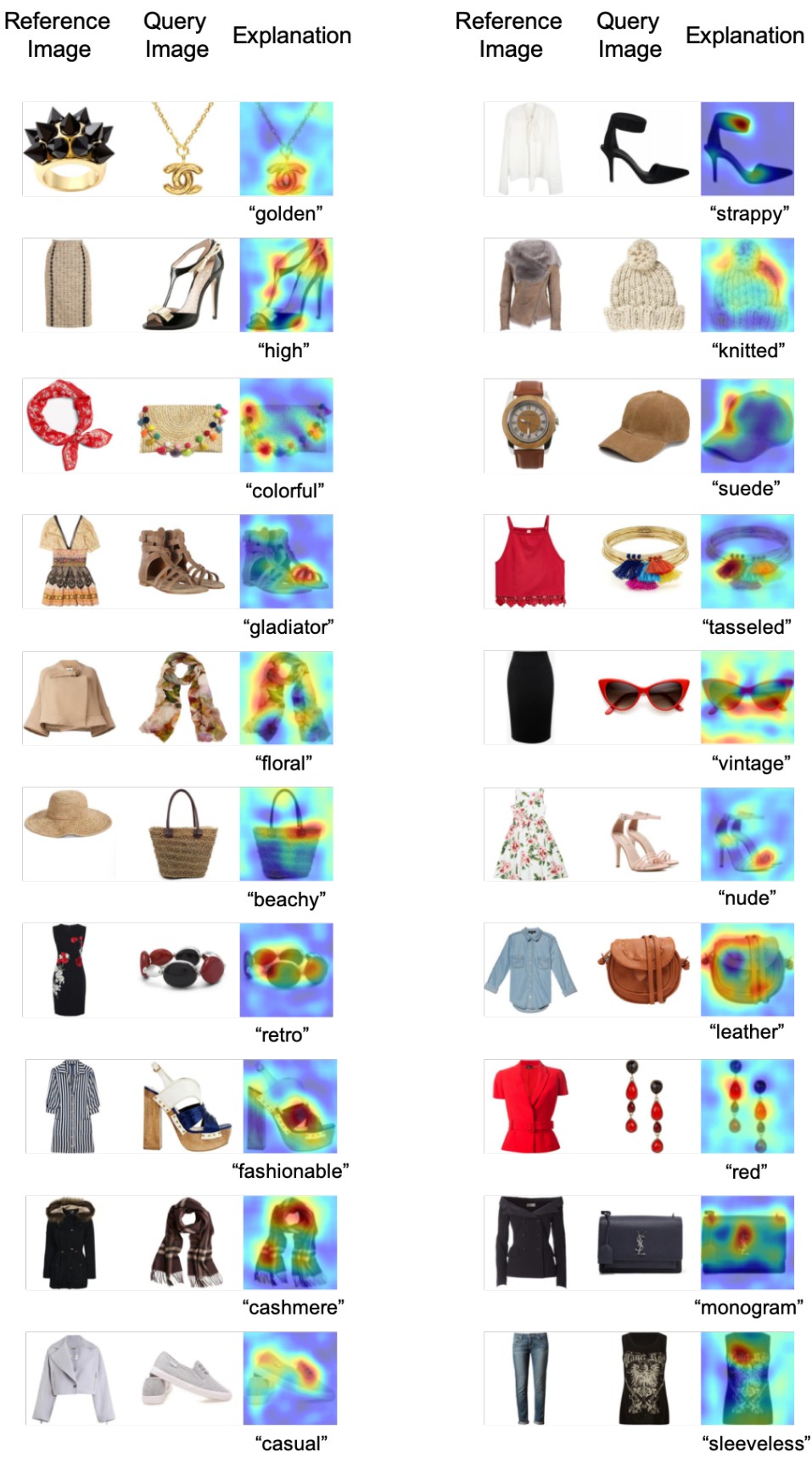

Figure 7: Additional qualitative examples of our SANE explanations on the Polyvore Outfits dataset.

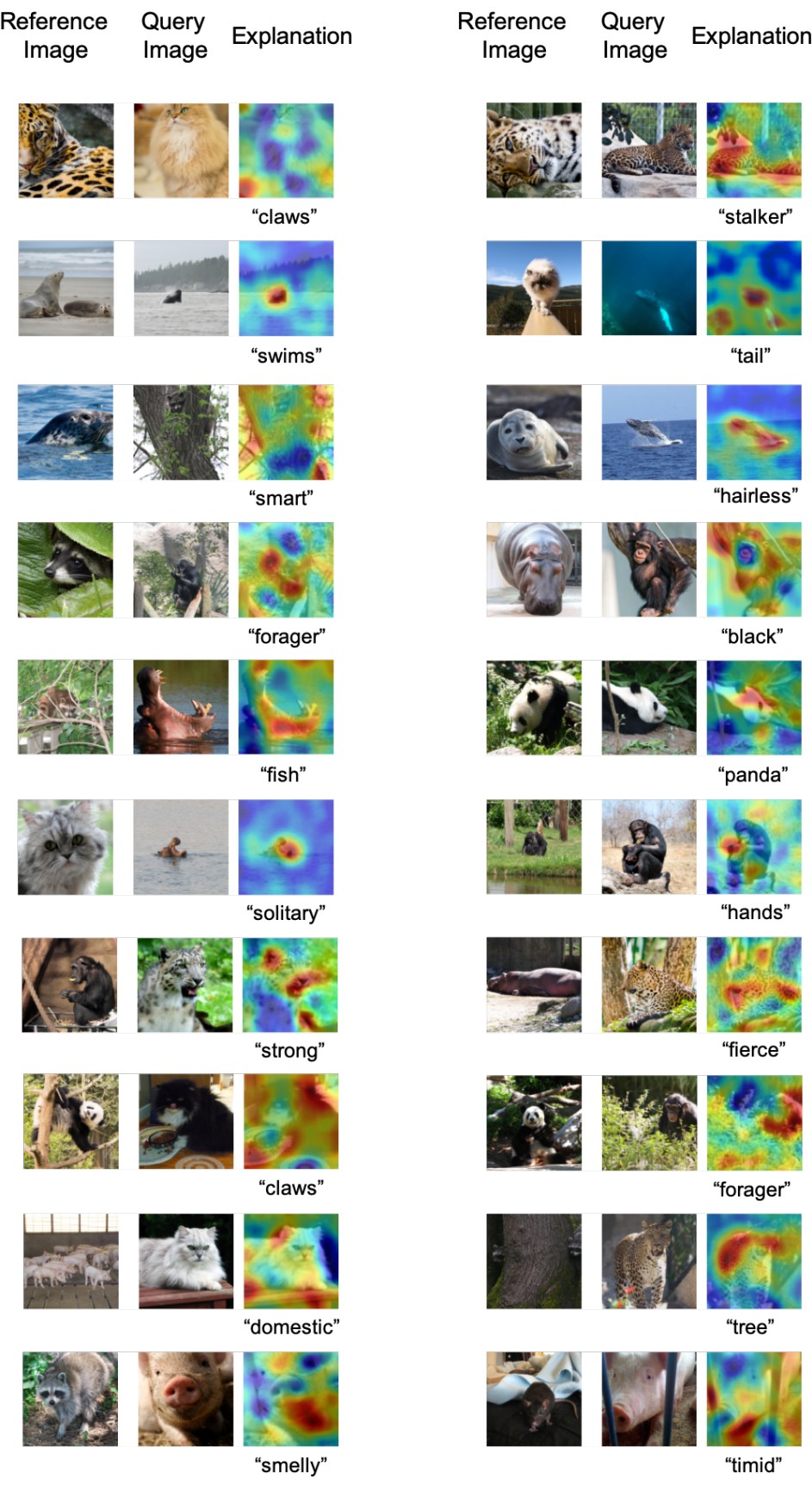

Figure 8: Additional qualitative examples of our SANE explanations on the AwA dataset.

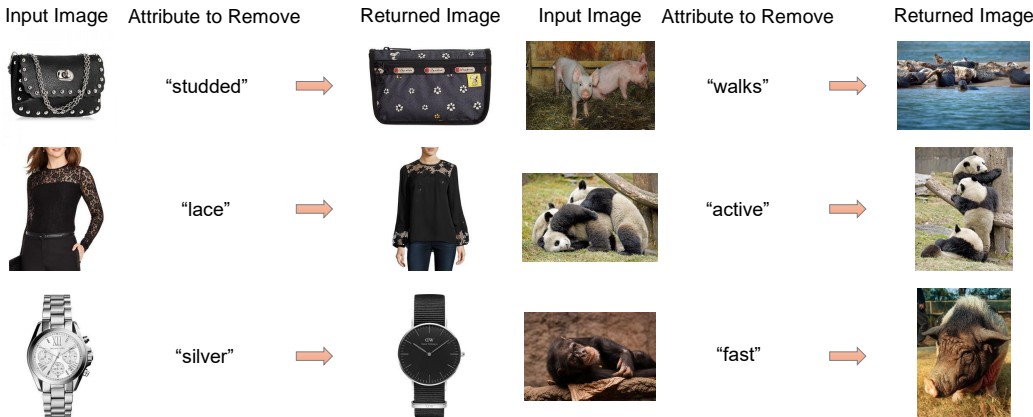

Figure 9: Examples of the attribute deletion process used to evaluate how good an attribute is as an explanation. We measure the similarity of the input image and some reference image as well as between the returned image and the reference image. If a large change in similarity is measured then the attribute is considered a "good" explanation. If similarity stays about the same, the attribute is considered a "poor" explanation, *e.g.*, trying to remove "active" from the pandas on the right.

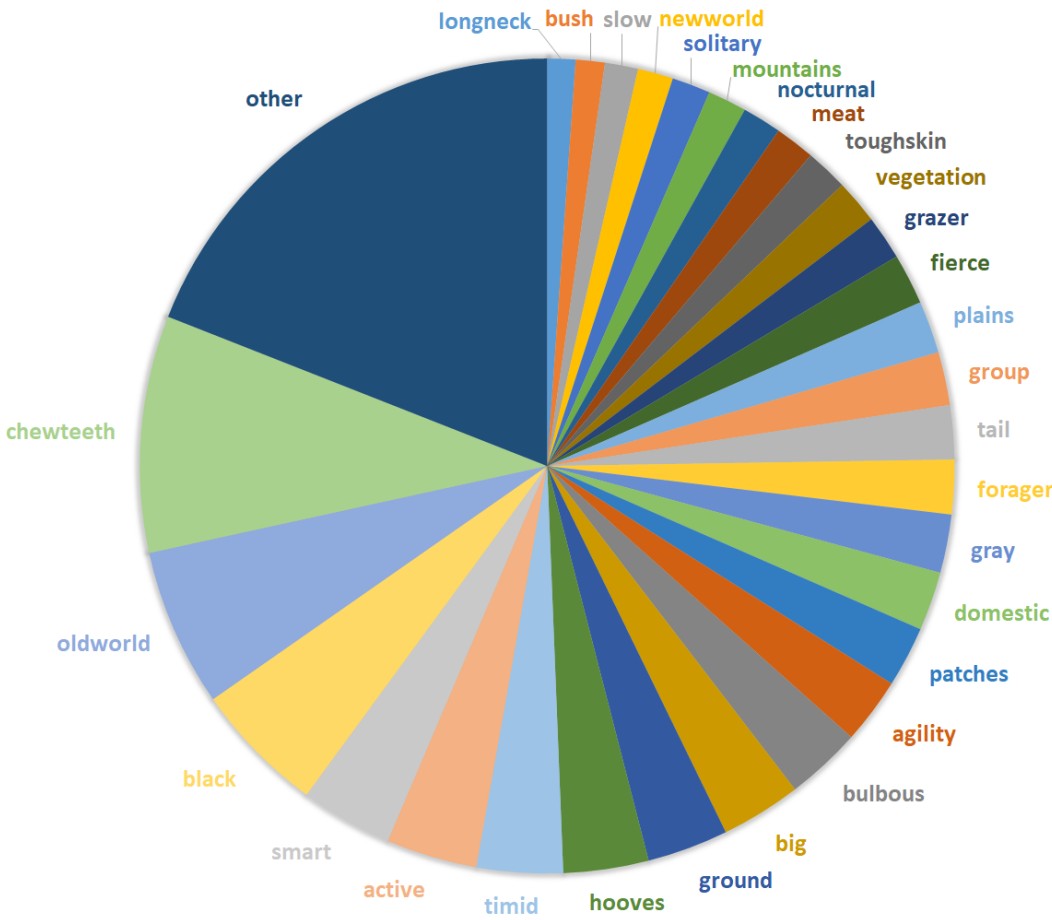

Figure 10: The likelihood each attribute in the AwA dataset was identified as the best attribute for an image pair on held-out data.

Table 4: Discovered attribute explanation performance comparison using the full SANE model.

| Attribute Types | Polyvore Outfits | | Animals with Attributes 2 | |
|---|---|---|---|---|
| | Insertion ($\uparrow$) | Deletion ($\uparrow$) | Insertion ($\uparrow$) | Deletion ($\uparrow$) |
| Random | 71.7 | 73.3 | 41.2 | 48.9 |
| Full Frame Discovery | 75.7 | 75.0 | 47.9 | 50.5 |
| Patch Discovery | 76.1 | 75.4 | 48.2 | 51.1 |
| Supervised Attributes | **76.8** | **76.2** | **48.9** | **51.6** |

be good attribute explanations. In Figure 10 we show the ground truth bias for the attribute detection task according to our metrics for the AwA dataset. Note, however, that this prior would change for a different image similarity model. For example, if the image similarity model was more biased towards colors, then we would expect to see the likelihood for "black," "brown," and "gray" to increase.

## C DISCOVERING USEFUL ATTRIBUTES

For datasets without attribute annotations, or those where the annotated attributes doesn't cover the extent of the visual attributes present in the dataset (*i.e.* there are many unannotated attributes) we propose a method of discovering attributes that are useful for providing model explanations. An attribute that is useful for explantions would commonly appear in the high importance regions of saliency maps. When generating saliency maps for a query image, if many reference images attend to the same region of the query image then it is likely they are all matching to it for similar reasons (*i.e.* there may be some attribute that they share which matches the query). Given this observation, we discover attributes using the following saliency-based procedure:

1. Obtain $K$ similar images for query image $q$ using k-NN.
2. Generate a saliency map over $q$ for each of the similar (reference) images.
3. Keep only those reference images which have their saliency peaks in the most common location (such as a unit square in a $7 \times 7$ grid) and pick top $N$ of them that have the highest similarity.
4. For each reference image, generate its saliency map with $q$ and crop a $30 \times 30$ patch around the peak saliency region in the reference image.
5. Upsample all the generated patches to full image resolution and get their embeddings.
6. Cluster the patches produced for multiple queries $q$. Each cluster represents an attribute. If multiple patches were extracted from an image and they got assigned to different clusters, this image would be labeled with multiple attributes.

Figure 11a illustrates the clustering produced by this procedure for a set of queries from Polyvore Outfits dataset.

To evaluate this approach we compare it to randomly assigning images to clusters and to clustering based on their own embeddings, disregarding the saliency of image regions (Figure 11b). Saliency-based attribute discovery works best among the three unsupervised methods for Polyvore Outfits data, but full-frame clustering outperforms it for the AwA dataset (Table 4). We suspect the full frame clustering works better for AwA since it considers the background more than the patch-based method (Polyvore Outfits image's typically have white backgrounds). In addition, our discovered attributes would likely be noisier due to the similarity model focusing on the background patches in some images as well. Although our initial results are promising, attempting to discover attributes useful for explanations warrants additional investigation.

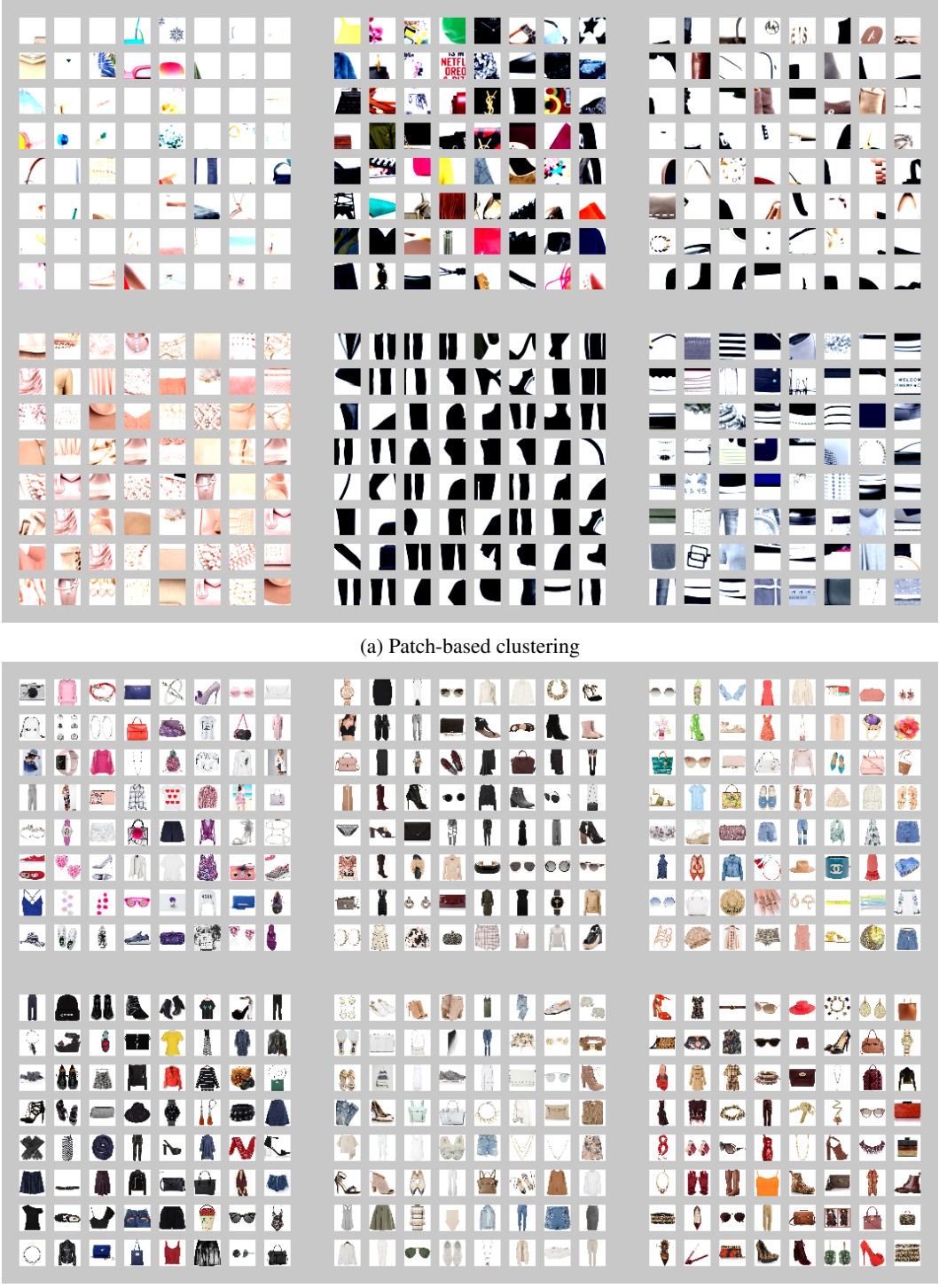

(a) Patch-based clustering

(b) Full-frame clustering

Figure 11: Six clusters defining the attributes for two approaches to attribute discovery. .
(a) Each image is assigned a list of clusters that have patches from this image. Clustering is performed on salient patches.
(b) Each image is assigned one of the clusters as an attribute. Clustering is performed on full-frame images.

