# OpenReview forum: "Why do These Match? Explaining the Behavior of Image Similarity Models"
_ICLR.cc/2020/Conference — Reject_

### Official Review · AnonReviewer1 · 2019-10-23
**Official Blind Review #1**

**Rating:** 6

**Review:**

I Summary
This paper proposes a novel method for image similarity models explanation, introducing Salient Attributes for Network Explanation (SANE). The method identifies attributes that contribute positively to the similarity score, thus explaining the important image properties, and pair them with a generated saliency map unveiling the important regions of the image. The method combines three major components:
- An attribute explanation model
- A saliency map generator where three "black box" algorithms are tested (sliding window, RISE, and LIME) and one "white box" (Mask)
- An attribute explanation suitability prior is computed by the weighted combination of the TCAV scores of an attribute, its confidence score and the matching of its activation map with the generated saliency map

Using the saliency maps as supervision for the attribute activation maps seems to improve attribute explanations. The obtained explanations help users understand the model's predictions and build trust.


II Comments

Overall the paper is well written and presents an interesting method for explaining image similarity models. However, from a writing perspective, it can be hard to follow as the paper lacks story-telling as to why such or such methods were chosen/implemented.

1. Content
- While this work is conceptually interesting, the technical novelty and contributions don't stand out as much as they could. What is the context in improving image similarity explainability? I believe examples in industry or medical could be found to highlight the story of the paper. Why a method is used over another? (TCAV, Mask etc, what lead to this choice?)
- In 2. Related work, Saliency-based explanations, the paper refers to white-box models but does not offer explanations as to why Mask is chosen over other methods (gradcam, guided backprop etc).
- In 3.3 TCAV is mentioned, as far as I know, the method works with concepts as images against random images. Here attributes are used as the concepts, how are the random counterparts selected? Moreover, the section on TCAV should be in the related work, whereas how it is used for this specific case would be described in 3.3.
- In eq 4, â is mentioned but s is used.
- In 4.2 there is a small user study to verify if the explanations were useful, the study is a nice addition, I really like this kind of results! It would be even more interesting if it compared the results with other baselines.
- The "discovering attributes" part in the appendix is promising, this is something that could be referred to in the conclusion.

2. Writing
- Intuitive and well-described explanations are given in most paragraphs with examples (3.2, Manipulating similarity scores, the button example) which give a good understanding of the problem. This led to a better comprehension of the challenges.
Small typos, did not impact the score:
- section 3. l 7 explanation -> explain
- section 4.2, results, l 5 effects -> affects

III Conclusion
The idea is interesting and seems to yield good results, especially in the appendix with the discovering attributes methods. The paper could sell itself a little better with more context/applications where it could be used.

**Experience Assessment:**

I have read many papers in this area.

**Review Assessment: Checking Correctness Of Derivations And Theory:**

I assessed the sensibility of the derivations and theory.

**Review Assessment: Checking Correctness Of Experiments:**

I assessed the sensibility of the experiments.

**Review Assessment: Thoroughness In Paper Reading:**

I read the paper at least twice and used my best judgement in assessing the paper.

---

> ### Author Response · Authors · 2019-11-15
> **Response to R1**
>
> We have used many of your comments to improve our paper in our updated pdf.  Direct responses to questions are addressed below.
>
> - What is the context in improving image similarity explainability?  I believe examples in industry or medical could be found to highlight the story of the paper.
>
> Some example applications have been added to the introduction.
>
>
>
> - Saliency-based explanations, the paper refers to white-box models but does not offer explanations as to why Mask is chosen over other methods (gradcam, guided backprop etc).
>
> Mask has shown to perform better than many other white-box alternatives like gradcam and guided backprop on many tasks.
>
>
>
> - TCAV is mentioned, as far as I know, the method works with concepts as images against random images. Here attributes are used as the concepts, how are the random counterparts selected?
>
> TCAV is now mentioned in the related work. The random images are selected from those which are not annotated with the target concept.

---

### Official Review · AnonReviewer2 · 2019-10-24
**Official Blind Review #2**

**Rating:** 6

**Review:**

Overview/Contribution:
====================
The paper proposes an explanation mechanism that pairs the typical saliency map regions together with attributes for similarity matching deep neural networks. The authors tested their methods on two datasets, i.e. Polyvore Outfits (a clothing attributes dataset) and Animals with Attributes 2 (a dataset of attributes for animals).

Overall, the paper has merit to be accepted to the conference with the following strengths and weaknesses. I suggest to the authors to address the weaknesses pointed out to make the paper more stronger, especially adding few more attributes datasets such as person attributes datasets as noted below in the weakness section.

Strength:
========
- The paper is written clearly and is easy to understand. I have seen the additional results and visual comparisons in the supplemental material and it was useful, albeit a bit longer.

- Explanations have the potential to make decisions made by a deep neural model transparent to end users among other benefits especially for sensitive applications such as healthcare and security. Explaining decisions made by similarity matching models has many applications including person attribute recognition and person re-identification for surveillance scenarios [1]. So, in this respect, this paper is relevant to the target audience.

- There is a bit of confusion between explanation and interpretation of decisions made by deep neural network models in the explainable AI literature and in most cases the two are used interchangeably. Hence, saliency maps are considered as explanation on their own by many. Combining saliency map based interpretations together with higher level concepts such as attributes has the potential to generate more realistic explanations of the decisions. The authors made this point at the second paragraph of the introduction.

- Fig. 1 (b) also is a clear example of the kind of explanations generated using a template with the key attribute in question accompanied by the visual saliency map interpretation.

- Fig. 2 clearly shows the overall proposed method and the attribute ranking based on the attributes explanation prior and the match between the saliency map and attribute activation maps.

- The attribute ranking and selection method of informative attributes using combinations weighted TCAV and cosine similarity between the attribute activation map and the generated saliency map is novel.

Weakness:
===========
- Applications of such a combined explanatory system don’t seem to be highly motivated in the introduction. I suggest the authors discuss more of the image similarity based applications and less on the discussion and heavy citation of generalized deep neural networks.


- The forms of the two loss components are both variants of l_{1} and l_{2} standard losses and they could be subject to issues with the standard variants of the l_{1} and l_{2} losses such as lack of translation and other transformation invariances. Hence, it would have been more useful to give the reasoning for the selection of the losses employed compared to other similarity and divergence based losses that are less sensitive to such variations.

- Similar to the above point, the choice of cosine similarity to compare match b/n attribute activation maps and saliency maps seem arbitrary. The method is described well but why cosine similarity was chosen in terms of its benefits compared to other similarity metrics is not that clear.

- Evaluation on more datasets such as person/pedestrian attributes datasets would have demonstrated the generalizability of the proposed method across multiple practical domains. As such, I would suggest the authors test their method on at least one person/pedestrian attributes dataset such as PETA, Market1501, etc.

- Although Fig. 1 (b) motivated a more practical high level explanation, in the results section, the attribute explanations are reduced to just the selected attribute that matched with the saliency well. Human-like concise attribute-based high level explanation just like the example given in Fig. 1 (b) would have made the paper stronger. Even if NLP is beyond the scope of this paper, a simple template based explanation that incorporated the selected/matched attribute would have been more effective.

- The results are too concise and a few ablation results on different losses etc. could have helped. There is too many qualitative results especially in the supplementary.

1) Bekele, E., Lawson, W. E., Horne, Z., & Khemlani, S. (2018). Implementing a Robust Explanatory Bias in a Person Re-identification Network. In Proceedings of the IEEE Conference on Computer Vision and Pattern Recognition Workshops (pp. 2165-2172).

**Experience Assessment:**

I have published in this field for several years.

**Review Assessment: Checking Correctness Of Derivations And Theory:**

N/A

**Review Assessment: Checking Correctness Of Experiments:**

I carefully checked the experiments.

**Review Assessment: Thoroughness In Paper Reading:**

I read the paper thoroughly.

---

> ### Author Response · Authors · 2019-11-15
> **Response to R2**
>
> - Applications of such a combined explanatory system don’t seem to be highly motivated in the introduction. I suggest the authors discuss more of the image similarity based applications and less on the discussion and heavy citation of generalized deep neural networks.
>
> We have made updates to the pdf to discuss this more.
>
>
>
> -  It would have been more useful to give the reasoning for the selection of the L1 and L2 losses compared to other similarity and divergence based losses.
>
> As discussed in the general comments, the L1 loss performs better than alternatives like sigmoid + binary cross entropy.  When comparing saliency map and attribute activation maps we use L2 loss, but these maps are compared for the same image.  Thus, these maps should align with each other exactly, making distance-based losses like L2 a good choice.
>
>
>
> - Similar to the above point, the choice of cosine similarity to compare match b/n attribute activation maps and saliency maps seem arbitrary. The method is described well but why cosine similarity was chosen in terms of its benefits compared to other similarity metrics is not that clear.
>
> As with the above point, since the maps being compared are from the same image, using distance-based metrics is ideal. Cosine similarity, in fact, is a very desirable similarity function since it effectively normalizes the features.
>
>
>
> - Evaluation on more datasets such as person/pedestrian attributes datasets would have demonstrated the generalizability of the proposed method across multiple practical domains. As such, I would suggest the authors test their method on at least one person/pedestrian attributes dataset such as PETA, Market1501, etc.
>
> We show that our approach performs well on two datasets from very different domains. Unfortunately, running additional datasets such as those referred to by the reviewer is not feasible within the rebuttal period.
>
>
>
> - A simple template based explanation that incorporated the selected/matched attribute would have been more effective.
>
> A template-based explanation is how we would expect our approach to be used in practice, and Figure 1(b) shows an example of how this would work. However, due to space constraints, for other qualitative results we showed the explanation attribute alone.
>
>
>
> - The results are too concise and a few ablation results on different losses etc. could have helped.
>
> As shown in the general comments and discussed earlier, we found other losses tend to hurt performance.

---

### Official Review · AnonReviewer4 · 2019-10-31
**Official Blind Review #4**

**Rating:** 3

**Review:**

This paper introduces SANE, a new approach for explaining image similarity models by combining a saliency map generator and an attribute predictor. In this way, the method is not only able to highlight what regions contribute the most to the similarity between a query image and a reference image, but also predict an attribute that explains this match. During training, SANE jointly optimizes the attribute prediction of the query image and maximizes the overlap of the saliency map of the image similarity and the attribute activations.

I think the paper addresses a very interesting problem that has been commonly overlooked. There are many recents works on the explainability of neural networks for images classification and other similar tasks, but very few have addressed this problem for image similarity. It is also novel and interesting the addition of an attribute predictor in the system which provides additional information that cannot be captured by the saliency map alone. Finally, the paper also does a big effort presenting a quantitative study of how SANE is able to explain similarity models.

However, I would also like to raise a couple of issues/questions regarding the method and its technical contribution:

- I would suggest the authors to include a brief explanation of the architecture used for the attribute predictor since it will help to understand how the attribute activation is computed. I am assuming that a Fully-Convolutional Neural Network is being used, where the output of the last convolutional layers has as many channels as the numbers of classes. Is this correct?

- Why using softmax + L1 loss to train the multi-attribute predictor? Aren't there other activations and losses better suited for multi-label classification, such as sigmoid + binary cross entropy loss, where there's no need to divide the ground-truth labels?

- In order to match image similarities with attribute descriptions the authors propose matching similarity saliency maps with attribute map activations. This is done by first computing a saliency map for the similarity between a query image and a reference image, computing the activation maps of the ground-truth attributes of the query image, then finding the attribute activation that best matches the saliency map, and finally minimizing the distance between the saliency map and the attribute activation using an L2 loss (cf last paragraph Section 3.1). My first question is: how is the best matching attribute match? I missed this explanation in the paper and, to my understanding, this is a very crucial step. My second concern is that I don't see why the attribute activation map should be matched with the similarity saliency map since not all the regions highlighted in the similarity map might describe the attribute. For example, if we're comparing two images  containing a jacket and both contain the attributes "zipper" and "black", the saliency map might highlight regions of the zipper and where the black color is present, but the activations of the attribute "black" should not be enforces to match the regions of the zipper. Could the authors explain the intuition behind this design choice?

- A final minor comment: as I mentioned before, the introdution of attributes in the explanation process is a very interesting contribution since they provide the user an explanation that is a step closer to a description in natural language. However, this comes at the price of needing attribute annotations at training. In order to overcome this problem, the authors suggest using an attribute discovery method when no attribute annotations are provided. My question therefore is: how are these automatically discovered attributes gonna be useful in order to provide a description, given that they are not associated with any word or concept?


Although the paper proposes a very interesting approach for explaining image similarity models, I also have some concerns that I think should be addressed before its acceptance. Therefore, my initial recommendation is weak reject.

**Experience Assessment:**

I have published in this field for several years.

**Review Assessment: Checking Correctness Of Derivations And Theory:**

N/A

**Review Assessment: Checking Correctness Of Experiments:**

I carefully checked the experiments.

**Review Assessment: Thoroughness In Paper Reading:**

I read the paper thoroughly.

---

> ### Author Response · Authors · 2019-11-15
> **Response to R4**
>
> - I would suggest the authors to include a brief explanation of the architecture used for the attribute predictor since it will help to understand how the attribute activation is computed. I am assuming that a Fully-Convolutional Neural Network is being used, where the output of the last convolutional layers has as many channels as the numbers of classes. Is this correct?
>
> Yes, this is correct, the pdf has been updated accordingly.
>
>
>
> - Why using softmax + L1 loss to train the multi-attribute predictor? Aren't there other activations and losses better suited for multi-label classification, such as sigmoid + binary cross entropy loss, where there's no need to divide the ground-truth labels?
>
> As discussed in our general comments, this is because softmax + L1 loss performed better.
>
>
>
> - My first question is: how is the best matching attribute match? I missed this explanation in the paper and, to my understanding, this is a very crucial step.
>
> At test time, the attribute explanation is selected using Eq. (4).  During training, we compute the loss function when supervising the attribute explanation maps with the saliency maps using Eq. (2).
>
>
>
> - My second concern is that I don't see why the attribute activation map should be matched with the similarity saliency map since not all the regions highlighted in the similarity map might describe the attribute. Could the authors explain the intuition behind this design choice?
>
> Our hypothesis is that the regions identified as important by a saliency map should be able to be explained by an attribute, and so the most prominent attribute at explaining the match should have an attribute activation map that is close to the saliency map. The intuition is that, typically, the most explanatory attribute for a match would be some salient property for the query image that dominates over others, and we find that to be empirically true - for instance, looking at the qualitative results in the appendix, we can see that the saliency maps are often well-localized, highlighting specific regions (e.g., the heel of a pair of high-heels). Even though there may be some cases where we expect the loss to be noisy, overall we found it improved performance. Thus, we can infer the saliency maps do follow our intuition much of the time, and that our hypothesis appears to be valid ( i.e., the high saliency regions can be described by an attribute).
>
>
>
>
> - How are automatically discovered attributes gonna be useful in order to provide a description, given that they are not associated with any word or concept?
>
> One could produce a human-interpretable label for these regions by showing the clusters to a human annotator and asking them to label them.  This would still be vastly more efficient than asking for complete attribute annotations for each individual image, and the attributes that are collected would exactly match those that would be important for explanations.

---

> > ### Author Response · Authors · 2019-11-15
> > **Addition**
> >
> > - ...the saliency map might highlight regions of the zipper and where the black color is present...
> >
> > As you noted, a saliency map might represent more than one attribute. We evaluated the performance of the top ranked attribute, but one could return the top K attributes using our model. We didn’t do this because attributes are often very strongly correlated and there is no generally accepted procedure for accounting for correlated attributes in the score.

---

### Author Response · Authors · 2019-11-15
**General Comments**

We thank the reviewers for their time and insightful comments.  Reviewers found that our paper addressed an interesting problem (R4) and introduced an interesting model (R4, R1) that has many applications (R2).

Multiple reviewers asked about our choice of loss functions, e.g.,  why softmax + L1 was used for our attribute recognition loss rather than alternatives like sigmoid + binary cross entropy. This is because softmax + L1 loss performed better in our experiments (e.g. sigmoid + binary cross entropy loss got 69.8/70.5 insertion/deletion while softmax + L1 got 71.7/73.3 on Polyvore Outfits).  This is likely due, in part, to the fact that sigmoid + binary cross entropy makes independent predictions for the presence of each attribute, whereas softmax + L1 loss trains a model where the attribute scores are calibrated so that relative scores of the attributes for an image are more meaningful. Since our task is to select which attribute is most relevant as an explanation, having calibrated scores is important.  That said, we still saw similar performance gains using SANE over the baselines even when using sigmoid + binary cross entropy, even though it worked worse overall than our approach.

Additional questions are responded to directly to each reviewer.

---

### Decision · Program_Chairs · 2019-12-19

**Decision:**

Reject

**Comment:**

This submission proposes an explainability method for deep visual representation models that have been trained to compute image similarity.

Strengths:
-The paper tackles an important and overlooked problem.
-The proposed approach is novel and interesting.

Weaknesses:
-The evaluation is not convincing. In particular (i) the evaluation is performed only on ground-truth pairs, rather than on ground-truth pairs and predicted pairs; (ii) the user study doesn’t disambiguate whether users find the SANE explanations better than the saliency map explanations or whether users tend to find text more understandable in general than heat maps. The user study should have compared their predicted attributes to the attribute prediction baseline; (iii) the explanation of Figure 4 is not convincing: the attribute is not only being removed. A new attribute is also being inserted (i.e. a new color). Therefore it’s not clear whether the similarity score should have increased or decreased; (iv) the proposed metric in section 4.2 is flawed: It matters whether similarity increases or decreases with insertion or deletion. The proposed metric doesn’t reflect that.
-Some key details, such as how the attribute insertion process was performed, haven’t been explained.

The reviewer ratings were borderline after discussion, with some important concerns still not having been addressed after the author feedback period. Given the remaining shortcomings, AC recommends rejection.